# IgG Antibody Responses to Epstein-Barr Virus in Myalgic Encephalomyelitis/Chronic Fatigue Syndrome: Their Effective Potential for Disease Diagnosis and Pathological Antigenic Mimicry

**DOI:** 10.3390/medicina60010161

**Published:** 2024-01-15

**Authors:** André Fonseca, Mateusz Szysz, Hoang Thien Ly, Clara Cordeiro, Nuno Sepúlveda

**Affiliations:** 1Faculty of Sciences and Technology, University of Algarve, 8005-139 Faro, Portugal; a49406@ualg.pt (A.F.); ccordei@ualg.pt (C.C.); 2CEAUL—Centre of Statistics and its Applications, Faculty of Sciences, University of Lisbon, 1749-016 Lisbon, Portugal; 3Faculty of Mathematics & Information Science, Warsaw University of Technology, 00-662 Warsaw, Poland; mateusz.szysz.stud@pw.edu.pl (M.S.); hoang_thien.ly.stud@pw.edu.pl (H.T.L.)

**Keywords:** biomarker discovery, disease pathogenesis, autoimmunity, antigenic mimicry, machine learning

## Abstract

*Background and Objectives:* The diagnosis and pathology of myalgic encephalomyelitis/chronic fatigue syndrome (ME/CFS) remain under debate. However, there is a growing body of evidence for an autoimmune component in ME/CFS caused by the Epstein-Barr virus (EBV) and other viral infections. *Materials and Methods:* In this work, we analyzed a large public dataset on the IgG antibodies to 3054 EBV peptides to understand whether these immune responses could help diagnose patients and trigger pathological autoimmunity; we used healthy controls (HCs) as a comparator cohort. Subsequently, we aimed at predicting the disease status of the study participants using a super learner algorithm targeting an accuracy of 85% when splitting data into train and test datasets. *Results:* When we compared the data of all ME/CFS patients or the data of a subgroup of those patients with non-infectious or unknown disease triggers to the data of the HC, we could not find an antibody-based classifier that would meet the desired accuracy in the test dataset. However, we could identify a 26-antibody classifier that could distinguish ME/CFS patients with an infectious disease trigger from the HCs with 100% and 90% accuracies in the train and test sets, respectively. We finally performed a bioinformatic analysis of the EBV peptides associated with these 26 antibodies. We found no correlation between the importance metric of the selected antibodies in the classifier and the maximal sequence homology between human proteins and each EBV peptide recognized by these antibodies. *Conclusions:* In conclusion, these 26 antibodies against EBV have an effective potential for disease diagnosis in a subset of patients. However, the peptides associated with these antibodies are less likely to induce autoimmune B-cell responses that could explain the pathogenesis of ME/CFS.

## 1. Introduction

The clinical manifestation of myalgic encephalomyelitis/chronic fatigue syndrome (ME/CFS) is typically a post-exertional malaise upon minimal physical and mental effort, a persistent fatigue that is not alleviated by rest, together with other symptoms related to neurologic, autonomic, and immunologic systems [1,2]. Several pathological mechanisms have been proposed to explain the origin of the disease and its progression over time [1,3,4,5,6]. Among these mechanisms, deleterious autoimmunity mostly driven by viruses is gaining traction in the literature [7,8,9]. SARS-CoV-2 is the newest causative agent of ME/CFS, as some long-COVID patients also comply with the diagnostic criteria for this disease [10,11,12,13]. However, the Epstein-Barr virus (EBV) along with other herpesviruses remain the usual suspects for causing ME/CFS and now long-COVID [14,15,16,17]. EBV is a particularly interesting virus given the growing evidence that, in some patients with ME/CFS, it enhances T follicular helper differentiation and promotes the formation of abnormal germinal centers that are essential for the generation of long-lived plasma cells and high-affinity antibodies [18]. The cause of these altered immune activities was hypothesized as an increase in activin A and IL-21 serum levels stimulated by EBV deoxyuridine triphosphate nucleotidohydrolase (dUTPase) in these patients [18]. EBV has also a strong potential for antigenic mimicry with human proteins, especially the EBNA1 protein, which contains highly repetitive glycine-alanine motifs [19,20]. This potential for eliciting autoimmunity has motivated serological investigations in patients with ME/CFS to identify key pathological EBV antigens and peptides [21,22,23,24]. However, these efforts did not lead to the identification of specific anti-EBV antibody signatures with a high accuracy in distinguishing patients from healthy controls (HCs).

These disappointing findings are typically explained by: (i) a very heterogenous clinical population; (ii) the presence of selection bias when recruiting patients; and (iii) the possibility of misdiagnosis where cases suspected of suffering from ME/CFS are actually genuine patients with another disease with a known cause [25,26]. An alternative explanation is an inadequate choice of the anti-EBV antibodies under analysis [27].

To avoid this problem, a recent study performed a large screening of IgG antibody responses to more than 3000 EBV peptides in patients with ME/CFS and HCs [28]. In a subsequent study on the same data, antibody responses to two peptides (EBNA4_0529 and EBNA6_0070) were identified as candidate biomarkers for the subgroup of ME/CFS patients whose disease started with an infection. However, these antibody responses were included in simple statistical models based on linear relationships between the antibodies (i.e., covariates) and the disease status. Therefore, the previous study could have failed to detect alternative antibody responses with more complex statistical relationships with the disease status. Also, the same study did not evaluate possible problems concerning data overfitting.

The present paper aims at re-analyzing the same dataset with the objective of using a machine learning approach, where the above analytical limitations could be tackled. We have also re-evaluated the role of eventual molecular mimicry between EBV and human antigens in the pathogenesis of ME/CFS.

## 2. Materials and Methods

### 2.1. Study Participants

Given that this study is a re-analysis of previously published data, the reader is recommended to consult the description of the original study in the respective reference [28]. In brief, 92 patients with ME/CFS were recruited from the Charité outpatient clinic for immunodeficiencies at the Institute of Medical Immunology in the Charité Universitatsmedizin Berlin, Germany. Fifty-four of these patients reported an acute infection at the beginning of their disease symptoms. The remaining patients (*n* = 38) reported not knowing their disease trigger or a disease trigger other than an infection. Fifty self-reported HCs were recruited from the staff of the same clinic.

Age and gender distributions of the ME/CFS cohort as a whole or divided into its two subgroups were matched with the ones of the HC cohort. See the corresponding analysis in Ref. [29].

### 2.2. Basic Description of Serological Data

The serological dataset under analysis is publicly available (see Supplementary File of Ref. [28]). In a nutshell, the serological dataset was generated by a seroarray that measured the signal intensities induced by individual IgG antibody responses to each one of the 3054 EBV peptides. These peptides were derived from 14 EBV proteins: BALF2, BALF5, BFRF3, BLLF1, BLLF3, BLRF2, BMRF1, BZLF1, EBNA1, EBNA3, EBNA4, EBNA6, LMP1, and LMP2. The peptides had a length of 15 amino acids (15-mer) and they could overlap within the same protein.

To denote each peptide, we used the protein name and its starting position within the corresponding protein, using the reference strain AG876. When the peptide name included *, it referred to the starting position of the reference strain B95-8.

### 2.3. Statistical Analysis for Predicting the Disease Status

#### 2.3.1. Dividing the Dataset into Train and Test Sets

Before conducting any analyses, the original dataset was divided into train and test subsets using a 9:1 ratio while maintaining the proportions of ME/CFS patients with subgroups, and the HCs. This ratio was approximately the optimal splitting ratio when applying a linear regression model that explains the variability of the data at the cost of 81 covariates [30].

#### 2.3.2. Ranking Antibodies by Their Importance for Predicting the Disease Status

We performed an initial step where we ranked the antibodies according to their importance in discriminating ME/CFS patients from HCs (Appendix A). In this step, we estimated 2500 random forests (RFs) using different hyperparameters: the number of trees (100, 500, 750, 1000, 2000), the number of features that could be used to split each node (fifty randomly generated values between 1 and 100), and the minimal node size (1, 2, 3, …, 10). These hyperparameters were modified in each run through a grid approach. In each run, the mean decrease in the Gini index was used to determine the importance of each antibody in predicting the disease status. After the 2500 runs, the importance value was determined by calculating the mean for each antibody and sorting them in descending order, identifying the most to the least essential antibodies for disease prediction.

In general, the Gini index measures the inequality of a given probability distribution. In the context of RF, the mean decrease in the Gini index is a measure of how each covariate contributes to the homogeneity of the nodes and leaves in the resulting RF. In this scenario, a higher mean decrease in the Gini index indicates that a given antibody is essential for the respective classification.

#### 2.3.3. Individual Statistical and Machine Learning Methods for Predicting the Disease Status from the Anti-EBV Antibodies

We applied 5 statistical techniques to predict the disease status based on the anti-EBV antibodies: elastic-net logistic regression (GLMNet), random forest (RF), support vector machine (SVM), linear discriminant analysis (LDA), and extreme gradient boosting (XGBoost). These methods were chosen due to their capacity for capturing different data patterns. On the one hand, GLMNet and LDA are based on linear combinations of the antibody values for predicting the disease status. On the other hand, RF, SVM, and XGBoost are particularly appropriate for detecting non-linear relationships between a set of covariates and the outcome. It is worth noting that probit regression could have been chosen. However, the probit and logistic regression models usually provide similar results due to their symmetric link functions. Regression models based on alternative link functions (e.g., log link) were also excluded from this analysis because there was no computational implementation for pooling their different disease predictions.

We increased the number of the most important antibodies to be included in each of the above 5 statistical techniques.

#### 2.3.4. Construction of Final Models for Predicting the Disease Status by Assembling Predictions from Individual Models

For a given number of antibodies, the results from the 5 individual classifiers were combined by the super learner (SL) algorithm, which assigns different weights to each individual classifier estimated for the same data [31]. Under generic assumptions, this algorithm typically improves the prediction of an outcome when compared to the accuracy of the predictions generated by each model individually.

The accuracy of the resulting classifier was evaluated by the proportion of individuals correctly classified using the ROC01 criterion [32]. This criterion dictates that the optimal accuracy is the one generated from a cut-off in the estimated classification probabilities that minimizes the distance between sensitivity/specificity to the perfect classification scenario (i.e., both sensitivity and specificity equal to 1).

We started our SL-based analysis with the two most important antibodies as the respective features/covariates. Every time the data from a new antibody response were added to the SL-based classifier (and its subclassifiers), we calculated the Spearman correlation coefficient R and removed highly correlated antibody responses (R > |0.8|), as done elsewhere [33]. This step was conducted in order to avoid redundancy and multicollinearity. We kept adding new antibody responses until we reached the maximum number of 100 antibody responses. The best classifier was the SL-based classifier with the lowest antibodies reaching the target accuracy of 85% in both the train and test sets; this accuracy is regarded as the optimal value for classification problems [34].

The above analysis was performed to compare the cohorts of all ME/CFS patients, ME/CFS patients with reported infectious disease triggers, and ME/CFS patients with non-infectious or unknown disease triggers against HCs.

### 2.4. Bioinformatic Analysis to Test the Importance of Antigen Mimicry in Predicting the Disease Status

When we found an SL-based classifier with the target accuracy in both train and test datasets, we then performed protein-protein alignments between the EBV peptide associated with each selected antibody and the human proteins included in the RefSeq reference protein database [35], as available in the National Centre for Biotechnology Information (https://blast.ncbi.nlm.nih.gov/, accessed on 1 August 2023). The quality of the alignments was based on the E-score statistic [36]. In our analysis, we focused on the maximal E-score associated with the alignments obtained for each EBV peptide under analysis. Subsequently, we calculated the Spearman non-parametric correlation coefficient between the importance of each selected antibody for disease prediction and the respective maximal E-score of the peptide associated with that antibody. We also calculated the respective 95% confidence interval. The same analysis was repeated using the human proteins included in the RefSeq non-redundant (nr) protein database.

### 2.5. Statistical Software

The statistical analyses were performed in the R software version 4.3.0 using the following packages: caret for multicollinearity analysis [37], OptimalCutpoints to obtain the accuracy based on the ROC01 criterion of each predictive model [32], pROC for the AUC estimation [38], ranger to perform random forest [39], and SuperLearner for the SL-based analysis [40].

## 3. Results

### 3.1. Construction of a Predictive Model to Distinguish All ME/CFS Patients from HCs

A comparison between all 92 patients with ME/CFS and the 50 HCs was carried out to develop a classifier that could predict the disease status of these study participants. The train dataset was composed of forty-five HCs and eighty-three ME/CFS patients, while the test dataset comprised five HCs and nine ME/CFS patients.

The overall average antibody importance distribution is presented in Figure 1A as a density plot. Our results showed that the overall average antibody importance was around 0.018, with the antibody against EBNA6_0066 being the most important (0.36). Furthermore, seven out of the ten topmost antibodies are associated with peptides belonging to the family of the Epstein-Barr nuclear antigen (EBNA) proteins.

It is worth noting that the levels of antibodies against EBNA1_430—a peptide with a potential molecular mimicry with the human Anoctamin-2 protein [41,42]—were similar in both ME/CFS patients and HCs (Appendix A). Consequently, they only had an average importance of 0.010 (ranked in the 1866th place of the most important antibodies). Therefore, this finding suggested a negligible role of these antibodies in predicting disease status.

In the train subset, the target accuracy of 85% was already achieved by an SL classifier including only two antibodies (Figure 1B). The corresponding sensitivity and specificity were close or equal to 1). This finding resulted from the high accuracy of the RF irrespective of the number of antibodies used as features (Figure 1C).

In the test subset, the accuracy estimates fluctuated around 50% and were at best 64%, using an SL classifier including 36 antibodies as features. This poor performance was explained mainly by the low sensitivity of the classifiers (Figure 1D). Hence, the target accuracy of 85% was not achieved for the overall dataset, largely due to poor performance in predicting ME/CFS patients in this data subset.

### 3.2. Construction of a Predictive Model to Distinguish ME/CFS Patients with Non-Infectious or Unknown Disease Triggers from HCs

We then compared the 38 ME/CFS patients with non-infectious or unknown disease triggers to the 50 HCs. This time, the overall average antibody importance was around 0.012. The most important antibodies were EBNA1_0595, EBNA1_0530, and EBNA6_0400, with the mean importance of 0.13, 0.12, and 0.11, respectively (Figure 2A). Once more, seven out of the ten topmost important antibodies recognized antigens from the EBNA protein group. In line with the analysis based on the whole cohort of ME/CFS, the antibodies against EBNA1_430 had an average importance of 0.005, which translated into a poor importance ranking (2384th place) among all the antibodies.

In the train subset, an SL classifier based on the top three antibodies (EBNA1_0595 EBNA1_0530, and EBNA6_0400) reached an accuracy of 94%, a value higher than the target accuracy of 85% (Figure 3B). In contrast, the same classifier reached only 78% in the test subset (Figure 2B). Such a value was the best accuracy that could be achieved for this analysis.

In the test subset, the target accuracy of the top three antibodies SL classifiers was not achieved by a relatively modest sensitivity (Figure 2C). Hence, this analysis suggested that these EBV antibodies were unable to discriminate this subset of ME/CFS patients from the HCs with high sensitivity and high specificity.

### 3.3. Construction of a Predictive Model to Distinguish ME/CFS Patients with a Putative Infectious Disease Trigger from HCs

We finally conducted a comparison between the 54 ME/CFS patients who reported an infection at their disease onset and the 50 HCs. This time the antibodies recognizing the EBNA4_0529, EBNA3_0139, and EBNA3_0577 antigens had the highest importance values in the RF (0.26, 0.25, and 0.24, respectively; Figure 3A). Eight of the ten topmost important antibodies belonged to the group of EBNA proteins. Once again, the antibodies against EBNA1_430 had a low average importance (0.009) and a poor ranking (1447th place) in terms of predictive importance.

In this analysis, we found an SL classifier including 26 antibodies that could reach an accuracy above the target value of 85% in both train and test datasets (99% and 90%, respectively; Figure 3B). These antibodies were associated with antigens derived from nine different EBV proteins: BALF2 (*n* = 2), BALF5 (*n* = 3), BLLF1 (*n* = 1), BMRF1 (*n* = 1), EBNA1 (*n* = 2), EBNA3 (*n* = 4), EBNA4 (*n* = 4), EBNA6 (*n* = 7), and LMP2 (*n* = 2). Twenty-two out of the twenty-six selected antibodies had increased levels in this subset of ME/CFS patients compared to the HCs (Figure 3C). The estimated classifier had a sensitivity of 100% and a specificity of 80% (Figure 3D). The corresponding AUCs for both the train and test datasets were 1.00 and 0.88, respectively (Figure 3E).

Note that an SL classifier including 42 antibodies predicted disease status almost perfectly in both train and test datasets (Figure 4B). However, this perfect classification was achieved at the cost of approximately 2.5 antibodies per study participant.

### 3.4. Testing the Importance of Antigen Mimicry on Disease Prediction Using a Bioinformatic Approach

The final analysis aimed at testing the hypothesis whether the EBV peptides associated with the above 26 antibodies selected in the SL classifier could explain the pathology of ME/CFS via a mechanism of molecular mimicry. Under this hypothesis, we expected a positive correlation between the importance of each antibody selected and the best alignment score between the associated peptides and human proteins.

The best alignment score per peptide varied from 22.7 (BALF5_0025) to 32.9 (EBNA6_0070) when conducting the protein alignments in the human RefSeq reference protein dataset. In the case of EBNA6_0070, the highest alignment scores were associated with alignments against the Homeobox-9A (HOXA-91) and adrenergic receptor alpha (ADRA1B) proteins (Figure 4A). The peptide with the second-best alignment was EBNA6_0488. In this case, this peptide had an extensive sequence homology with the human proteins CCCTC-binding factor (CTCF) and adipocyte enhancer-binding proteins (AEBP1) (Figure 4B).

When we analyzed the average importance of these 26 antibodies against the best alignment scores of the respective peptides, we found a slight negative association between these two quantities, but this association was not statistically significant (R = −0.161, 95% CI = (−0.473; 0.185); Figure 4C). We obtained the same lack of correlation using the non-redundant protein database (R = −0.018, 95% CI = (−0.355; 0.330); Appendix A). Hence, these data did not support the hypothesis that antibodies recognizing important EBV peptides for disease prediction had a high potential for cross-reactivity with human proteins.

## 4. Discussion

### 4.1. General Comments

This paper has demonstrated the recurrent difficulty of finding anti-EBV antibodies that could be used as general markers of ME/CFS. Similar difficulty was encountered in studies on IgG antibodies against common pathogens [43] or multiple peptides derived from different herpesviruses [21]. In our case, such a difficulty was mainly due to patients who reported a non-infectious disease trigger or did not know their disease trigger. Therefore, stratifying patients by the respective disease trigger was sufficient to make specific EBV antibody signatures emerge for patients with an infectious disease trigger. A more detailed discussion of the utility of stratifying patients by their disease trigger can be found in our previous works [24,29]. A more general discussion of the issue of patient stratification can be found in Jason et al. [44].

In the present work, the best-case scenario was obtained for the patients with an infectious disease trigger where a set of 26 EBV-related antibodies led to good accuracy in both train and test datasets. This finding is not surprising, given that our previous study also led to a similar conclusion but using a different statistical methodology [29].

Similar large-scale antibody screening was performed on patients suffering from multiple sclerosis (MS), a disease strongly correlated with EBV infections. In one of these screenings, the levels of the 26 identified antibodies were not significantly different in patients with MS when compared to healthy donors [45]. This finding suggests that our 26-antibody signature is specific to ME/CFS patients. However, in one of the largest studies of MS, the levels of antibodies related to EBNA1_0005, EBNA3_0577, EBNA4_0566, EBNA6_0025, EBNA6_0488, and EBNA6_0752 peptides were significantly elevated in MS patients [46]. This previous finding is in line with the recurrent observation that some patients with ME/CFS share the same symptomology [47,48] and antibody alterations with patients with MS [49].

### 4.2. Clinical and Diagnostic Implications

The identification of multiple elevated anti-EBV antibodies linked to the cohort of ME/CFS patients with an infectious disease trigger suggests that treatments such as immuno-adsorption, rituximab, and cyclophosphamide could be deployed to treat this clinical group specifically. Interestingly, the clinical value of these three treatment options was at the heart of a recent meeting entirely dedicated to ME/CFS research [50].

In general, immuno-adsorption is an aphaeretic procedure that removes specific proteins (e.g., antibodies) from a patient’s plasma. This treatment has already been tested successfully in a small cohort of ME/CFS patients with an infectious disease trigger whose autoantibodies against adrenergic receptors were elevated [51,52]. The same treatment was recently tested in long COVID ME/CFS patients, with similar successful outcomes [53]. Given that immuno-adsorption induces a significant depletion of total IgG, we speculate that the clinical benefit of this treatment comes from the removal of not only autoantibodies against adrenergic receptors from patients but also the EBV antibodies here identified.

The administration of rituximab, an anti-CD20 monoclonal antibody, aims at inducing a slow depletion of hypothetical autoreactive B cells in ME/CFS patients [3]. This depletion also has the advantage of removing EBV-infected B cells. At the same time, the depletion of the peripheral B-cell pool induces the renewal of the B-cell pool by a healthy one coming from the bone marrow. The initial clinical trials on the deployment of this drug to ME/CFS treatment were promising [54,55]. However, similar results were not observed when the clinical trial was scaled to a larger cohort of ME/CFS patients [56]. A possible explanation for this finding is the use of rituximab to treat all patients irrespective of their disease. A better treatment strategy is to use this drug only on patients with an infectious disease trigger, mainly those who reported an infectious mononucleosis at the disease onset.

Cyclophosphamide is an immunosuppressive drug that was already tested, with promising results, in ME/CFS patients [57]. This drug generally acts on both T and B cells [58]. Interestingly, early studies on cyclophosphamide demonstrated a strong suppressive effect on antibody formation in animal models [59,60]. In addition, there is evidence of a decrease in the B-cell numbers after cyclophosphamide administration in patients suffering from rheumatoid arthritis [61]. Given that EBV can trigger both T- and B-cell responses and the antibody production requires the activation of the CD4+ T helper cells recognizing a given antigen, this drug is expected to have a higher clinical benefit than rituximab due to a broader suppression of the adaptive immune responses towards potentially pathogenic EBV peptides. In light of this perspective, it would be interesting to confirm whether CD4+ T helper cells from the same cohort of ME/CFS patients can be activated by the same EBV peptides associated with the identified 26-antibody signature. If such an activation occurs, it provides a strong rationale for using cyclophosphamide instead of rituximab to treat these patients.

Our results suggest that 26 antibodies could be used jointly as a diagnostic tool for suspected cases who reported an infection at the onset of their symptoms. This is of great clinical value, given that more than 64% of ME/CFS patients report an infectious disease trigger [24,62,63]. However, further studies should be conducted to understand the effect of disease duration, the infectious agent, and other factors on sensitivity. On the other hand, the hypothetical diagnostic potential of these 26 antibodies requires going beyond the routine use of ELISA for a single antibody testing. In this regard, the practical use of a 26-antibody diagnostic tool demands the use of multiplex and other high-throughput serological platforms in which these antibodies could be quantified simultaneously, as done for testing malaria exposure [64,65]. If so, the 26-antibody diagnostic tool could be conducted in reference clinical centers dedicated to ME/CFS, where high-throughput serological platforms are available.

Unfortunately, we could not find any significant EBV antibody signatures for patients who did not know or report a non-infectious disease trigger. The same negative result was obtained in our previous analysis of the same data [29]. It is worth noting that non-infectious disease triggers might be, among others, pregnancy, surgery, personal and work-related stress, or exposure to chemicals [24,63]. These triggers seem not to have as direct and strong an impact on the immune system as infections do. Hence, it is perfectly conceivable that patients with non-infectious disease triggers have pathological mechanisms that are more related to dysfunctions of the body’s systems other than the immune one. As such, alternative approaches (e.g., metabolomics, DNA methylation, or genetics) should be sought to tackle the pathogenesis of patients who did not know their disease trigger or reported a non-infectious one, and then to discover the respective biomarker.

### 4.3. EBV Antigenic Mimicry and Its Putative Role in ME/CFS Pathogenesis

#### 4.3.1. Replication of Previous Finding on EBNA6_0070 Peptide

Among the peptides recognized by the selected antibodies, EBNA6_0070 had the highest sequence homology with a human protein. This antigen had already been discovered and amply discussed in our previous study [29]. The replication of this finding using a different statistical approach provided further support for the hypothetical role of this peptide in ME/CFS. However, in contrast with our previous work, antibodies against this candidate antigen were not among the top most important antibodies for disease prediction. This result suggests that the potential pathological effect of this EBV antigen via molecular mimicry is not as so straightforward as our initial study suggested.

To resolve this question, one could purify IgG antibodies against this EBV peptide from ME/CFS patients and then transfer them to recipient humanized mice. Alternatively, one could synthesize the EBNA6_0070 peptide artificially and then inject it into humanized mice. One can then measure the motor activity of these mice after these challenges. Classical measures of motor activity are distance travelled, voluntary wheel running, or time standing still [66]. Suppose this peptide or the respective recognizing antibodies are indeed in the causal pathway of ME/CFS. In that case, one should observe a significant reduction of motor activity in these challenged mice over the time course of the experiment. It is worth noting that animal models are already in use to understand the role of antibodies on fibromyalgia [67], a co-morbidity of many ME/CFS patients. Animal models also provide proof-of-principle evidence of a hypothetical pathological mechanism for ME/CFS [18]. Efforts were also conducted to develop a mouse model for ME/CFS, using lipopolysaccharide challenges [68,69]. However, it is unclear how these models really mimic the main disease symptoms, especially post-exertional malaise.

#### 4.3.2. EBNA6_0488 Peptide and the Antigenic Mimicry with CTCF and AEBP1

The EBNA6_0488 peptide had the second-highest sequence homology with human proteins. This homology was related to two possible human 10-mer peptides belonging to CTCF and AEBP1. The former protein is a master transcription factor due to its more than 50,000 possible binding sites and its role as a chromatin barrier element [70,71]. In addition, the level of this transcription factor is inversely correlated with the levels of DNA methylation [71]. CTCF in partnership with cohesion molecules is also important in many immunological pathways, such as the interferon gamma production in Th1 cells and the establishment and maintenance of regulatory T cells in visceral adipose tissue and skeletal muscle [72,73]. In this scenario, we speculate that the increased quantity of antibodies against EBNA6_0488 results in a cross-reactive antibody response to the CTCF peptide, thus reducing the abundance of this transcription factor. This putative reduction could lead to altered gene expression and DNA methylation patterns, and abnormal immunological processes, including the maintenance of deleterious autoimmunity in check. This speculation is in line with findings from altered gene expression and DNA methylation profiles in ME/CFS patients [74,75,76,77]. As an extreme case, one study identified more than 12,000 CpG sites with altered DNA methylation levels in patients with ME/CFS compared to HCs [78]. Immunological abnormalities are also reported by many studies in ME/CFS patients (reviewed in Refs. [8,79]). An alternative interpretation is that the increased levels of antibodies against EBNA6_0488 resulted from a putative CTCF overexpression during the disease progression in the cohort of ME/CFS patients with an infectious disease trigger. An overexpression of this transcription factor could be the result of a stress-induced response to restore homeostatic equilibrium within cells. However, altered gene expression was not reported for CTCF by any gene expression studies published so far. This negative reporting could be explained by not performing any patient stratification when analyzing data from these studies.

With respect to AEBP1, this protein is a ubiquitous transcriptional repressor involved in the regulation of adipogenesis, mammary gland development, inflammation, macrophage cholesterol homeostasis, and atherogenesis [80]. Interestingly, mutations on the AEBP1-encoding gene were implicated with the onset of Ehlers-Danlos syndrome (EDS) [81,82]. Patients with EDS hypermobility type can also receive a diagnosis of ME/CFS [83]. On the other hand, patients with a diagnosis of ME/CFS also show EDS as a co-morbidity [84]. In fact, the presence of EDS in a suspected case of ME/CFS has not been considered as an exclusionary condition for the respective disease diagnosis [85]. However, genome-wide association studies of ME/CFS did not report any genetic markers located in the AEBP1 gene [86,87,88,89,90]. In this scenario, antibody responses to EBNA6_0488 with the potential of being cross-reactive with AEBP1 should alter the regulation of biological processes where this protein is involved. In particular, the deficient regulation of inflammatory processes is particularly relevant for ME/CFS, given the general idea that established ME/CFS translates into a persistent low-grade inflammatory process in patients [6]. Given that endothelial dysfunction is also observed in patients with ME/CFS [91,92,93], such a dysfunction could result from damaged endothelial cells via persistent low-grade inflammation in response to EBNA6_0488 mimicking an AEBP1 peptide. Hence, the identification of this molecular mimicry brings an unexpected link between EBV and AEBP1. As alluded above for CTCF, current gene expression studies did not highlight AEBP1 at the top of the proteins with the most significant differential abundance between patients with ME/CFS and HCs. The lack of patient stratification is once again a possible reason for not detecting an altered abundance of AEBP1 in ME/CFS patients when compared to HCs.

Interestingly, the maximum sequence homology of the peptides recognized by the 26 selected antibodies and human proteins was not associated with the importance of the same antibodies in disease prediction. Moreover, antibodies against EBNA1_430, which contains a peptide mimicking a peptide from the human Anoctamin-2 protein [41,42], had low importance in predicting ME/CFS patients. These results suggest that potential molecular mimicries due to antibody reactivity between EBV and human antigens have a minor role in the underlying pathological mechanism in such a subset of patients. However, we cannot rule out the possibility of a potential mimicry based on the three-dimensional molecular structure of the respective peptides, but not at the level of their amino-acid sequence. We cannot also rule out that molecular mimicries based on sequence homology might be elicited by peptides from EBV proteins other than the ones evaluated in this study. This might be the case of two EBV peptides from BPLF1 and BHRF1 proteins that were able to elicit an immune response by self-reactive T-cell clones derived from patients with MS [94].

### 4.4. Interpretation of the Findings under the Lens of the Danger Theory

An interesting perspective on the above results can be given by the so-called danger theory [95]. The theory is based on the premise that the immune system is activated by danger or damage signals sent by infected (or stressed) cells to the immune system. These danger signals are independent of the intrinsic nature of antigens (self or non-self) seen by the immune system. As a corollary, autoimmune responses and autoimmune diseases are then understood as unintended consequences of persistent danger signals that ultimately include chronic presentation of multiple self and non-self antigens. This explains why chronic and low-grade infections by herpesviruses are among the most documented triggers of autoimmune diseases. In this scenario, the theory exactly predicts the lack of correlation between the importance of the selected antibodies in predicting ME/CFS and the degree of molecular mimicry between the EBV peptides and human proteins.

The basic question of applying the danger theory to ME/CFS pathogenesis lies in understanding which danger signals are at the core of the disease. According to the original proponents of the danger theory, general danger signals are the heat shock proteins (HSP), the vasoactive intestinal polypeptide (VIP), and the cytokines TNFα and IL1β, among others [96]. A brief discussion about some of these danger signals in the context of ME/CFS is given below; a more comprehensive discussion of this topic will be conducted in the near future.

### 4.5. Potential Danger Signals in ME/CFS Pathogenesis

HSPs are highly conservative proteins in nature and are produced in response to many different cellular stresses. In theory, antigens derived from these proteins were thought to belong to the so-called immunological homunculus, a limited set of dominant self antigens that allow the immune system to have a picture of the self [97]. However, there is no consensus on whether HSPs are indeed signaling danger or are simply key regulatory and resolution elements of a stress or immune response [95,98]. This alternative interpretation of the functional role of HSPs might explain the lack of consistency in HSP-related responses across studies where patients with ME/CFS and HCs were challenged with physical exercise [99,100,101]. In addition, antibodies against endogenous and microbial HSP65 peptides were at the same level in patients with ME/CFS and HCs, with the exception of a higher seroprevalence to an HSP65 peptide derived from *Chlamydia pneumoniae* in the former [102].

VIP is a neuromodulator present in the gut and the anterior chamber of the eye [96]. On the one hand, it can activate dendritic cells [103] (thus its suggestion as a potential danger signal). Conversely, the binding of VIP to its receptor in immune cells also leads to anti-inflammatory actions [104]. In this line of thought, a loss of tolerance to VIP, other vasoactive neuropeptides, or their receptors was hypothesized to be at the genesis of ME/CFS [105]. However, a follow-up study showed an elevated expression of VPACR2—the VIP receptor—in immune cells and an increased frequency of the regulatory Foxp3+CD4+ T cells in ME/CFS patients in comparison with HCs [106]. Given that the generation of these regulatory cells can be induced by VIP [107], this finding is more in line with this neuromodulator being a mediator of regulation in the context of ME/CFS.

TNFα and IL1β are two classical pro-inflammatory cytokines. According to a systematic review [108], it was found that 20–25% of the studies reported elevated levels of these cytokines in patients with ME/CFS when compared to HCs. However, the same systematic review did not perform a meta-analysis of the published data. Therefore, it is unclear whether the lack of significant findings related to these two cytokines results from insufficient statistical power due to reduced sample sizes used in the respective studies. It is worth noting that ME/CFS patients from Italy had a higher frequency of an allele variant associated with elevated levels of TNFα (rs1800629:G>A) [109]. However, this finding was not replicated by another study with German patients [110].

TNFα and IL1β are also known to bridge the adaptive and innate arms of the immune system via the so-called CD40/CD40 ligand (CD40L) pathway. In particular, CD40L and its mutations trigger different immunological signaling cascades on B cells [111] that might be important for the establishment of EBV latency and its reactivation. The fundamental question is to understand how TNFα, IL1β, and CD40L are balanced under normal and disease-related conditions. On the one hand, there is ample evidence that TNFα and CD40L influence the immunological activity of each other [112,113,114]. In the context of Crohn’s disease, TNFα can show anti-inflammatory activity by down-regulating the CD40/CD40L pathway [115]. This capacity might be an explanation for the observation that CD40L levels were significantly decreased in ME/CFS patients with shorter disease duration [116,117]. From this perspective, the increased levels of TNFα in ME/CFS might be seen as an anti-inflammatory response to an exacerbated immune response to an ongoing infection (caused by EBV or another virus) that initiated ME/CFS. On the other hand, IL1β is known to cooperate with CD40L to increase the production of pro-inflammatory cytokines and activate dendritic cells [118,119]. In light of the above evidence, it is an interesting research question to know the interplay between TNFα, IL1β, and CD40L at the early stages of ME/CFS.

## 5. Conclusions

In summary, this study provided a list of possible EBV peptides whose associated IgG antibody responses could be used in the diagnosis of suspected ME/CFS cases who reported an infection at their symptoms’ onset. Two of these peptides had a high sequence homology with human proteins, but the corresponding antibody responses were not the most important ones for disease prediction. This finding suggested that the role of EBV on eventual ME/CFS-related autoimmunity should be reconsidered under the lens of danger theory.

## Figures and Tables

**Figure 1 medicina-60-00161-f001:**
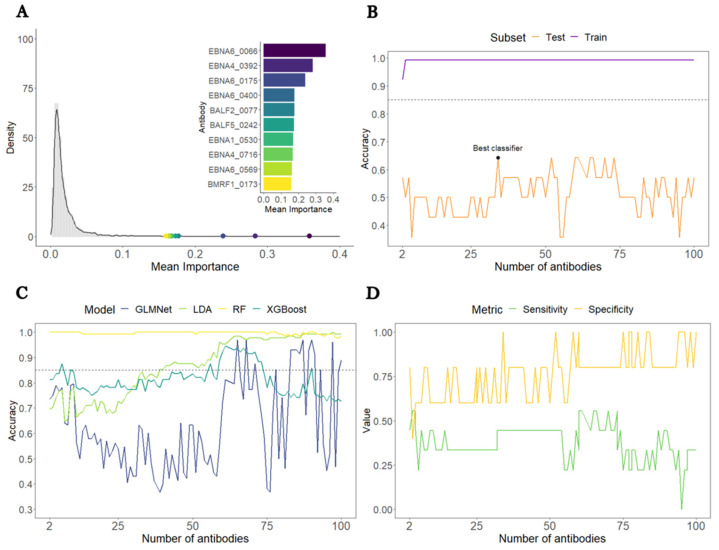
Analysis of all ME/CFS patients versus HCs. (**A**) Density plot of each antibody’s average importance distribution obtained by the RF with the top 10 most important antibodies highlighted. (**B**) Accuracy of the SL classifier in the train (purple) and test (orange) subsets as a function of the number of antibodies included. The black and blue horizontal dashed lines indicate 85% (target accuracy). The best classifier is highlighted with a black dot. (**C**) Accuracy of the different classifiers assembled by the SL in the train subset. (**D**) Sensitivity and specificity of the SL classifiers in the test subset as a function of the number of antibodies included.

**Figure 2 medicina-60-00161-f002:**
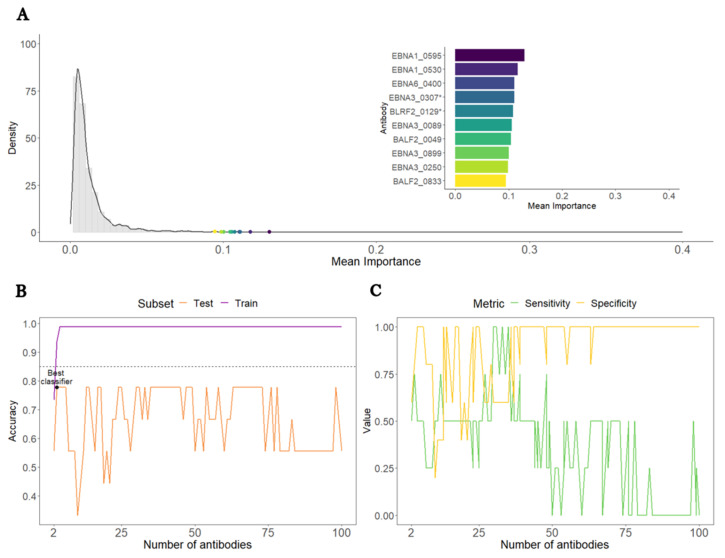
Analysis of ME/CFS patients with non-infectious or unknown disease triggers against HCs. (**A**) Density plot of each antibody’s average importance distribution obtained by the RF with the top 10 most important antibodies highlighted. (**B**) Accuracy of the SL classifier in the train (purple) and test (orange) subsets as a function of the number of antibodies included. The black and blue horizontal dashed lines referred to the 85% target accuracy. The best classifier is highlighted with a black dot. (**C**) Sensitivity and specificity of the SL classifiers in the test subset as a function of number of antibodies included.

**Figure 3 medicina-60-00161-f003:**
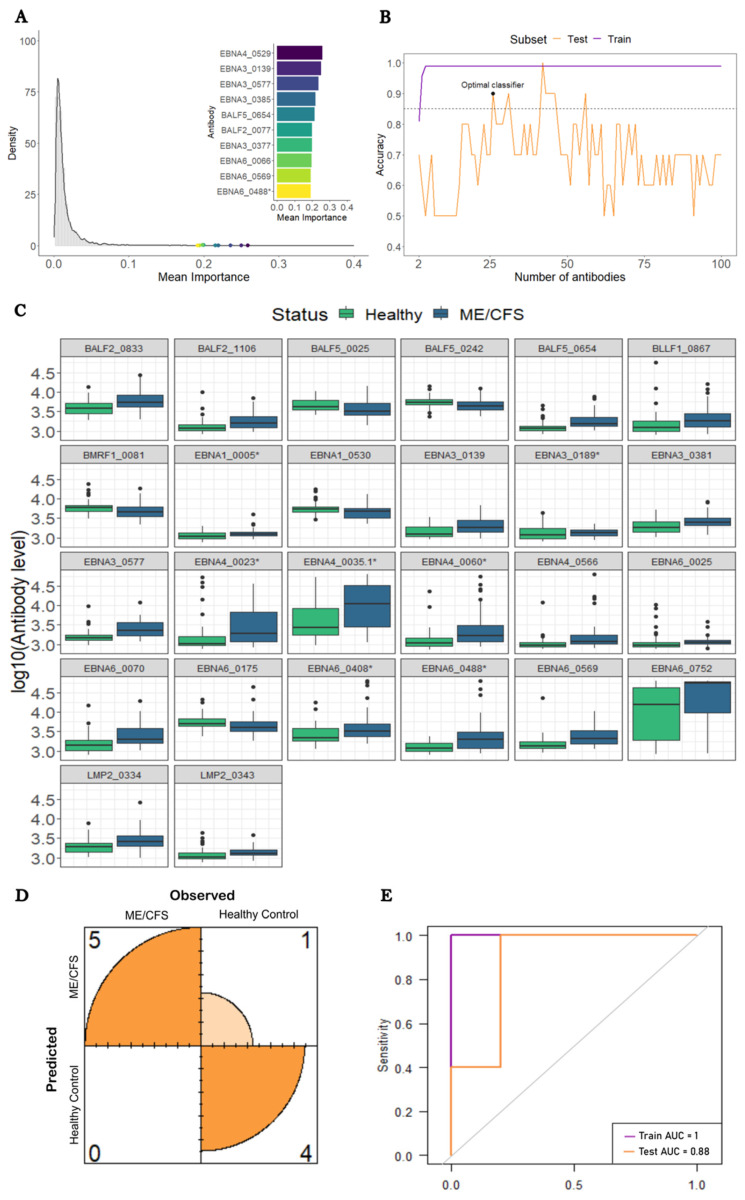
Analysis of ME/CFS patients with an infectious disease trigger against HCs. (**A**) Density plot of each antibody’s average importance distribution obtained by the RF with the top 10 most important antibodies highlighted. (**B**) Accuracy of the SL classifier in the train and test subsets as a function of the number of antibodies included. The black and blue horizontal dashed lines indicate the target accuracy of 85% (i.e., 0.85). The best classifier is highlighted with a black dot. (**C**) Boxplots of the log10-levels of the selected 26 antibodies in HCs and ME/CFS patients. (**D**) Confusion matrix concerning the optimal classifier performance on the test subset. (**E**) ROC curve of the optimal classifier for both train and test subsets.

**Figure 4 medicina-60-00161-f004:**
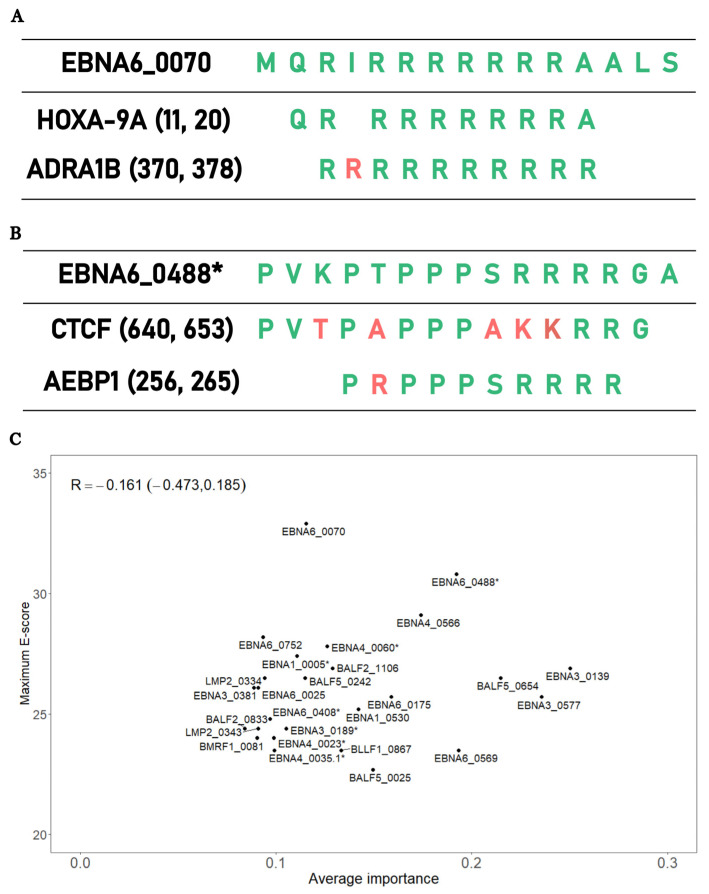
Bioinformatic analysis of the EBV peptides associated with the 26 antibodies for predicting ME/CFS patients with an infectious disease trigger. (**A**) Alignments between EBNA6_0070 and the human proteins HOXA-9A and ADRA1B_371 with the corresponding amino-acid coordinates within brackets. (**B**) Alignments between EBNA6_0488* and the human proteins CTCF and AEBP1 with the corresponding protein coordinates within brackets. (**C**). Scatterplot between the average importance of each EBV peptide and the maximum E-score alignment score with human proteins using the RefSeq reference protein database, where R is the Spearman correlation coefficient with the respective 95% confidence interval in brackets.

## Data Availability

The dataset is freely available as Supplementary Materials of Loebel et al. [28] (accessed on 1 March 2023). The R scripts are available without restriction at the following address: https://github.com/Publications/Fonseca_etal.

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
