# Peer review of "IgG Antibody Responses to Epstein-Barr Virus in Myalgic Encephalomyelitis/Chronic Fatigue Syndrome: Their Effective Potential for Disease Diagnosis and Pathological Antigenic Mimicry"

_medicina, 2024, doi:10.3390/medicina60010161_

Round 1
Reviewer 1 Report
Comments and Suggestions for Authors
The paper investigates the difficulty of finding anti-EBV antibodies as general markers for ME/CFS, emphasizing the heterogeneity of the patient population. Here are some critical comments and suggestions for improvement:
1. Clarity and Structure:
The paper lacks a clear structure, making it challenging for readers to follow the flow of information. Consider restructuring the paper with clear headings and subheadings to guide the reader through the different sections. The introduction could provide more context on ME/CFS, the significance of finding reliable markers, and the existing challenges in the field.
2. Repetition and Redundancy: The paper repeats the notion of patient heterogeneity and the difficulty of finding a generic disease marker multiple times. Consider consolidating this information to avoid redundancy and improve the overall coherence of the paper.
3. Statistical Analysis: The paper mentions statistical methodologies but lacks details on the methods employed. Provide more information on the statistical analyses, including the rationale behind the chosen methods and potential limitations.
4. In-depth Discussion of Results: The paper discusses specific antibodies associated with EBV peptides but does not delve into the clinical implications or potential applications of these findings. Provide a more in-depth discussion on how these antibodies might contribute to our understanding of ME/CFS or impact patient diagnosis and treatment.
5. Discussion of Limitations: The paper briefly mentions the difficulty in finding markers for patients who do not know their disease triggers. Discuss the implications of this limitation in more detail and explore potential strategies or alternative approaches to address this challenge.
6. Connection to Previous Studies: While the paper references a previous study, it could benefit from a more comprehensive review of existing literature on anti-EBV antibodies in ME/CFS. Discuss how the current findings align or differ from previous research.
7. Proposal for Further Research: The paper suggests performing functional studies to investigate the potential pathological effects of specific EBV antigens. Provide a more detailed proposal for these future studies, including the experimental design, methodology, and expected outcomes.
8. Conclusion: The conclusion should summarize the key findings, their implications, and potential avenues for future research. Avoid introducing new information in the conclusion and focus on reinforcing the study's main contributions.
9. Grammar and Style: Proofread the paper for grammatical errors and improve the writing style for clarity and coherence.
10. In-text Citations: Ensure all claims and statements are supported by appropriate in-text citations. Cross-reference claims with the cited literature to strengthen the scientific validity of the paper.
Addressing these points should help enhance the overall quality and impact of the paper.

Reviewer 2 Report
Comments and Suggestions for Authors The manuscript entitled "IgG antibody responses to Epstein-Barr virus in Myalgic Encephalomyelitis/Chronic Fatigue Syndrome: their effective potential for disease diagnosis and pathological antigenic mimicry" submitted by Fonseca et.al. discusses the role of autoimmunity caused the Epstein-Barr virus and other viruses in the Myalgic Encephalomyelitis/Chronic Fatigue Syndrome. The authors use a public data set for the IgG antibodies against the EBV peptides to use them as a predictive marker for the disease diagnosis. These data were compared against the healthy controls. Using the bioinformatics approach the authors were able to predict the disease status at 85% accuracy. The authors identified a 26-antibody classifier which could be used for the distinguishing the ME/CFS patients from the healthy individuals. The authors were able to show this to be accurate to the tunes of 90% using a test dataset. The authors found that there was no sequence homology between the EBV peptides and the human proteins which is recognised by these 26 antibodies which explains they are less likely to show autoimmune responses. The authors had a clear hypothesis and the approach used for testing the hypothesis is streamlined by set experimental design. The quality of data presented, and the statistical test are done appropriately to the best of my knowledge. The materials and methods section are complete and gives all the necessary information. Finally, the discussion is well written and discusses the possible mechanism and proposes further studies. However, there are some concerns which are as follows.In terms of the mechanistic approach do the author plan to check the levels of activin A and IL-21 in these patients which were tested. As there is growing evidence that these two are associated with a seropositivity for antibodies against the EBV and are critical for T follicular helper (TFH) cell differentiation and for the generation of high-affinity antibodies and long-lived plasma cells. The authors should explain this PMID: 35482424 and cite this in the manuscript. There are reports that the levels of CD40L is lower in the ME/CFS patients. This is an important molecule which helps in the generation of antibody diversity. The differential interaction between the CD40-CD40L results in the release of different cytokine profile and signalling which shapes the antibody release. the author should add these article PMID: 26079000,PMID: 32827854 and mention them in the discussion section. Also, the authors mentioned that the cytokines TNFa and IL1b in the line 408 is a pro-inflammatory cytokine and these are also induced by the CD40-CD40L signaling. Therefor the authors need to mention the above two articles in the discussion and refer them. Overall, the work done by Fonseca et.al. is commendable and adds to the necessary
